Maybritt Jill Alpes
Senior Researcher
Human Rights Centre
Ghent University
m.j.alpes@gmail.com

*Article submitted for peer review with*
*Migration Politics*

# Smuggling critique into impact:
# Research design principles for critical and actionable migration research

**Abstract:**

The article examines how academics can mobilize their epistemic resources to engage with justice claims able to challenge border violence. Many migration scholars would like to find ways to mobilize their knowledge to resist migrants' human rights violations. Despite increased focus on research impact, border violence is only increasing. On the one hand, policy makers do not act on scholarly recommendations that are highly critical, but not necessarily actionable. On the other hand, when scholarly recommendations are actionable, legal and policy changes do not necessarily result in meaningful improvements for refugees' and other migrants' dignity. As a result, there is a dichotomy between applied research that is not critical and critical research that is not actionable. Against this backdrop, this article explores how migration researchers can reclaim the meaning of impact and smuggle critique into the term. The article is based on auto-biographical explorations of what it means for an anthropologist to produce knowledge on migration from within law faculties and as policy officer and research consultant for human and refugee rights organizations. Based on this material, the article argues that migration scholars who seek justice should not produce more evidence, but rather take law seriously as a knowledge practice. The article develops three design principles for migration scholars who seek to resist in the short- and medium-term migration laws and policies that violate human right principles. First, build knowledge alliances with justice actors. Second, theorize knowledge needs in justice claims. Third, broker the validity of truth claims.

# Smuggling critique into impact:
## Research design principles for critical and actionable migration research

The stakes of responsible (migration) scholarship are high at times when the rise of right-wing extremism is shaking the core of democratic principles, including threats to the rule of law. As people are dying at borders at massive scales as a result of migration policies (Cuttitta and Last 2020), EU states simultaneously feed political narratives so as to deflect responsibilities for migrants' death, as well as lie outright about state practices to escape accountability (Davies, Isakjee, and Obradovic-Wochnik 2022; Aradau and Perret 2022). The European Commission, for example, continues to blame migrants' death on smugglers. And Greek police and coastguard officers purposefully destroy and prevent the production of evidence of its own practices at borders.[1]

Many migration scholars would like to find ways to mobilize their knowledge to resist migrants' human rights violations, starting with the right to life. Despite increased focus on research impact, however, border violence is only increasing. This dilemma raises questions about the potential of academic knowledge to critically and constructively interact with policy makers, legislators and decision-makers. On the one hand, policy makers do not act on scholarly recommendations that are highly critical, but not necessarily actionable. On the other hand, when scholarly recommendations are actionable, legal and policy changes do not necessarily result in meaningful improvements for refugees' and other migrants' access to life, dignity and rights. As a result, there is a dichotomy between applied research that is not critical and critical research that is not actionable.

The article examines how academics can mobilize their epistemic resources to engage with justice claims able to challenge border violence. Under what conditions can the meaning of 'impact' be reclaimed so that it contains the grains of meaningful critique? I define critical and actionable migration research as proposing constructive critique that can be translated into concrete actions in political and legal processes in the short and medium term. This definition is based on an understanding of critique as the contestation of "regimes of truth that govern us" (Lorenzini and Tazzioli 2020, 28) and of action as resistance to the making and implementing of migration laws and policies that violate human right principles. Critical and actionable migration research is relevant to political and legal categories, processes and temporalities, but is not necessarily driven or fully predetermined by it. Instead, critique most often requires a reframe of the assumptions of political and legal practitioners. In exploring possibilities for critical and actionable migration research, the article seeks to open up a grey space between policy-irrelevant (Bakewell 2008) and policy-driven research.

To smuggle critique into impact requires reflexive thinking about conditions of possibility for transformative knowledge encounters. To examine these conditions, I draw on the reflexive work of anthropologists of development. Here, anthropologists have distinguished several ideal types of how scholars interact with practice and its practitioners (Grillo 1985: 28-31, Olivier de Sardan 2008) (Lavigne Delville and Fresia 2018). First, "rejectionists" operate a radical critique that questions the epistemological foundations of aid and thus fundamentally

---

[1] [CPT/Inf (2020) 35] Page: 25-26. Available at coe-greece-cpt-report-march-2020.pdf (statewatch.org). Last accessed: 25 May 2023.

rejects the idea that anthropology could be useful for action. Rejectionists do not collaborate with practitioners. Second, "monotorists" operate with an operational critique aimed at improving action without questioning underlying frameworks. Monotorists are involved in the definition and implementation of projects. Third, "conditional reformers" believe that anthropologists have a useful role to play, provided that they are "fully aware of what involvement in that world means, ethically, politically and practically" (Grillo and Rew 1985, 31). They operate a comprehensive critique that is attentive to the critical reflexivity of practitioners themselves.

Transferring these ideal types to the field of migration politics, critical and actionable migration research comes from a conditional reformer positionality and is based on comprehensive critique. As such, critical and actionable migration research is not applied research, nor is it fundamentally void of theory. Taking seriously multiple epistemic sites (Aradau and Huysmans 2019), critical and actionable migration research does not assume that academia has a monopoly over knowledge and thus approaches practitioners as situated thinkers. Far from an essential characteristic, both scholars and practitioners can flexibly navigate between positionalities and types of critiques at various moments in times and in different contexts. Despite their daily work, some practitioners are actually very aware and open to even radical critique. And scholars can choose to mobilize different kinds of critiques depending on the respectively desires knowledge encounter.

The article is based on auto-biographical reflections on what it means for an anthropologist to produce knowledge from within law faculties, as well as policy officer and consultant for human and refugee rights organizations. From both positionalities, I have worked and reflected on the role of evidence in human rights advocacy and litigation. Policy advocacy refers to advocating for changes to laws, regulations, or government policies. Strategic litigation, also known as impact litigation or public interest litigation, refers to legal complaints in courts aimed at achieving structural changes beyond the direct resolution of an individual case. Advocacy and litigation are separate, yet connected domains. When human right defenders struggle to win political arguments, legal judgments can lend symbolic authority to claims on states. When judges have pronounced right violations, for example, the implementation of judgments still requires further advocacy by human right defenders. As such, political and legal processes are deeply intertwined.

Within a range of violent border practices, my auto-biographical reflections draw on experiences for the case of return practices and pushbacks. Since the EU Turkey statement, return practices are increasingly emmeshed with asylum procedures at European external borders. In pushbacks, migrants are forcefully pushed across borders in violation of the non-refoulement principle which prohibits the forcible return of refugees or asylum seekers to a country where they may face persecution or serious harm. To examine what happens after both returns and pushbacks, I did fieldwork on the African continent (Cameroon, Democratic Republic of Congo, Nigeria, Niger, Mali), at external European borders (Greece, Italy), as well as in the Middle East (Turkey, Lebanon).

I argue that migration scholars who want to seek justice should not produce more evidence. "Impact" is nothing to be "done" once the research is completed. Instead, producing critical and actionable migration research has to start at the research design phase. Critique can only be smuggled into impact if the why, what and how of a research project consider

conditions of possibility for transformative knowledge encounters. In a quest to resist in the short- and medium-term migration laws and policies that violate human right principles, I propose three research design principles for critical and actionable migration research. All three principles take seriously the knowledge practices of practitioners themselves.

Advocacy with legislators and government officials and litigation in Court are important sites of struggle for migratory justice as critique here can bring significant changes to the lives of refugees and other migrants in the short- and medium-term. If today's world is organized by nation states, then law can serve to hold states accountable, notably by limiting and questioning the legitimacy of the use of violence. Despite law's critical potential, resistance to epistemic and border violence requires a multitude of politics and complementary engagement strategies, including those who also focus on more radical and long-term transformations. Other scholars, for example, use theoretical debates to frame political options for grassroot movements, actively contribute to public debate in the media, seriously engage in learning conversations with students and pupils, seek out political changes in universities in a quest to decolonize knowledge production, collaborate with people who create school books or museum exhibitions, or seek to reach broader audiences through documentary films.

The mobilization of epistemic resources for justice claims is relevant also beyond migration studies. Violence in Gaza and transitional justice efforts in Syria, for example, are also marked by epistemic violence that begs for a critical reflection on responsible scholarship. At the same time, some universities also move to wanting to be "social change actors", adding new requirements and increasing work load and expectations for academic staff. These developments require critical internal debates about what social change is and what role universities can or should (not) play in this. If externally imposed and not internally reappropriated in meaningful ways, today's impact agenda can have detrimental consequences for academic freedom. Mindful of both post-truth politics and the neo-liberalization of academia, I hope this article will open safe discussion spaces that make it possible to go beyond critique on absolute terms, and towards a politics of care and responsibility for academic contributions to society.

After a sketch of the auto-biographical data and methodology, the article first explores how (migration) scholars can mobilize epistemic resources to smuggle critique into impact. Based on a theoretical discussion, I propose that both evidence and law constitute political resources. I illustrate these propositions through an auto-biographical exploration of how I have in the past mobilized epistemic resources to play different kinds of legal games. The article's second part explores conditions of possibility for transformative knowledge encounters. In doing so, I propose three design principles for migration scholars who seek to smuggle critique into impact: First, build knowledge alliances with justice actors. Second, theorize knowledge needs for justice claims. Third, broker the validity of truth claims.

**Auto-biography as method**

As an anthropologist, I have worked on migration laws and policies both inside and outside of academia. Inside of academia, I worked for nine years (2011- 2017, 2021- 2024) as a post-doctoral and senior researcher at the law faculty of respectively the VU Amsterdam and Ghent University. Outside of academia, I worked for five years (2015- 2020) respectively in different positions: as "research consultant" for human and refugee rights organizations, as "policy

officer" for Amnesty International, as "principal investigator" for a research project funded by the Open Society Foundation, and as "lead researcher" for a Syrian NGO in Lebanon. These different job titles gave me different degrees of autonomy during research design, fieldwork and analysis. The grant from OSF, for example, allowed me to design a research project about post-deportation risks from scratch. In another context, however, I was once faced with the request of a funder to deliver raw data, and not analysis. Here, I chose to resign after failed negotiations for more autonomy.

There are important differences in the why, what and how of research inside and outside of academia. While research outside of academia can be theoretically informed, for example, its objective is not to advance social science theories or academic debates in specific disciplines or fields. Nevertheless, the experience of this double career has resulted in productive overlaps in my case. Whenever possible, for example, I negotiated copyright agreements so that I could use fieldwork data from consultancies for follow-up publications in other venues and in my own name. And, some commissioned consultancies have resulted in prolonged collaborations, which in the end also informed the research design of two successful grant applications.

A relatively high degree of autonomy to shape my own research agenda outside of academia was possible in the aftermath of the European 2015 crisis of refugee governance. The perception of a migration crisis resulted in increased funding for civil society actors, including for research. Outside of academia, for example, I was funded for over a year of fieldwork in both familiar and new locations on three continents, as well as able allocate time to learn a new language, Arabic, from scratch. Nevertheless, the word 'consultant' sounds dirty in the realm of the social sciences. I remember a coffee conversation with a former university colleague where he stressed how NGOs were captured by state interests. At the time, he did not mention constraints of a neo-liberal academic university system, state capture of universities, crisis chasing in academia (Cabot 2019) and the subtle politics of EU funding schemes (Kalir and Cantat 2020).

This article project gives me an opportunity to reflect back on my own practice of producing and submitting evidence to policy makers and judges. When I engaged in these practices, I had no premeditated intention to subsequently use my action research as fieldwork data. In the absence of fieldnotes, I reviewed my different publications, events, fieldwork trips, including email conversations with grant-makers, lawyers and advocacy officers. In doing so, I focused in particular on moments of frustration, learnings and unexpected developments. In order to protect the respective organizations and actors I worked with, I will mobilize my auto-ethnographic data to mainly reflect on my own actions, including my mistakes.

**Smuggling critique into action: finding the sweet spot where resistance meets impact**
Migration studies today is marked by a lack of dialogue between conventional scholars basing themselves on Weber and critical scholars basing themselves on Marx and Foucault (Favell 2022). This lack of dialogue has also resulted in conversations on impact and resistance that are entirely disconnected from each other.

On the one hand, the quest for impact has led some social scientists in migration studies to engage in policy-driven research. Narrowing down the research objectives, policy-driven research risks reproducing the worldviews and priorities of policy makers and states

(Andersson 2018; Cabot 2016; Bakewell 2008), [2] including through damage-centered research (Lindberg 2024). Migration research here tends to be oriented more towards state actors. On the other hand, the desire for resistance has led other social scientists in migration studies to anchors politics in resistance struggles. Foregrounding root causes of border violence in racism and colonialism (Essed and Nimako 2006) (Sharma 2020), theory-driven research risks not connecting with the making and implementing of laws and policies in today's world. Migration research here tends to be oriented more towards migrant and refugee voices, as well as no-border movements.

Against this backdrop, I contend that resistance and impact do not have to be fundamental opposites. There is a sweet spot where resistance can meet impact. In order to bring together these disconnected and partially opposing aspirations, it is necessary to reflect on epistemological positions that make it possible to use knowledge for politics. Does impact in legal and political processes necessarily presuppose a positivist stance? Adopting a social constructivist approach, I propose that both evidence and law can be mobilized as resources to smuggle critique into impact.

Evidence as a political resource:
In a quest to go beyond the exclusively destructive nature of much critique in academia, the article puts forward that evidence can be a political resource to resist epistemic and border violence. Drawing on Medina, I understand epistemic violence as the forced de-legitimation and repression of certain possibilities of knowing, going hand in hand with an attempted enforcement of other possibilities of knowing (Medina 2013). At European borders, state authorities systematically stifle the capacity of refugees and human right defenders to draw attention to the empirical realities of border deaths, inhumane and degrading treatment and access barriers to asylum procedures. These practices of denialism and the offsetting of blame to migrants themselves have become so wide-spread that epistemic violence has arguably become inseparable from borderwork (Davies, Isakjee, and Obradovic-Wochnik 2022).

By putting forward that evidence can be a political resource to resist border and epistemic violence, I seek to embed research impact in a broader context of post-truth politics. 'Post-truth' politics is characterised by the dominance of guestimates, alternative facts and outright lies in public and political discourse (Kelly and McGoey 2018) and directly connected to the "decay of truth" (Kavanagh and Rich 2018; Kakutani 2018) in public debates. At times of post-truth politics, the question arises what responsibility and possibilities scholars have to mobilize their epistemic resources to move beyond no-harm ethics (Borofsky and De Lauri 2019; Stierl 2020), as well as about appropriate tactics. Critical studies of border violence do not only enable violence to be named, they can also unwittingly reify dominant representations without actually ending or at least mitigating violence (Lindberg 2024).

Embedding research impact into a broader context of post-truth politics implies for me two reverse acts of smuggling, notably critique into impact and impact into critique. First, impact is critical when it also seeks to influence how migration policies are framed. Evidence

---

[2] Even scholarship explicitly critical of state control can reproduce policy assumptions and relegate long-term historical continuities to the backstage when caught up in the rush of chasing the latest migration crisis (Cabot 2019).

can be a trojan horse, used to smuggle uncomfortable, but audible truth claims into legal and political processes. This smuggling act requires a balancing act and a flexible evolution between radical and comprehensive types of critique. Second, critique will only have impact when it contains propositions for feasible actions for legal and political actors. Evidence can be a political resource to translate selected contestations of regimes of truth into such actions. For example, when taking decisions, political and legal actors require knowledge that they can act on within the frameworks of their mandates. Legislators, government officials and ministers shape legal and political processes predominantly in the short and medium term. In doing so, critical and actionable migration research takes into account the temporal horizons of practitioners in political and legal processes.

By proposing to smuggle critique into impact and the reverse, I'm trying to account for social constructivist approaches to truth, but not be to be immobilized by Foucauldian conceptions of power and knowledge. Impact is a term mostly used by scholars with more positivist leanings (Mayne et al. 2018), resistance by scholars who are more in a post-modern camp (Medina 2017). Whilst expertise is a social construct that function as a technology of power (Kuus 2011), scholars can also choose to try and use or change how these social constructs operate in legal and political processes. Here, theories of how knowledge is used in policy-making can be of interest (Boswell 2008).

I propose that evidence can be a political resource if scholars build on the insights of social constructivist critiques of truth regimes in order to build constructive strategies for alternative pathways. Precisely because evidence has hardly informed post-2015 responses to the EU governance crisis (Baldwin-Edwards, Blitz, and Crawley 2019), knowledge can be produced and mobilized to challenge underlying assumptions and vested interests that in fact do drive the politics of policy-making. When strategically mobilized in this manner, evidence can be a political resource for epistemic resistance.

Medina has defined epistemic resistance as the use of "epistemic resources and abilities to undermine and change oppressive normative structures and the (..) functioning that sustains those structures" (Medina 2013). The critical potential of evidence here is not radical. Yet, resistance does not require a whole-scale rejection and can be based on subversion, too (Medina 2013).

Law as a political resource:
In a quest to go beyond the exclusively destructive nature of much critique in academia, the article puts forward that law can be mobilized as a political resource for constructive critique in migration politics. While law has been criticized to protect the status quo and to favour those who are more powerful (Israël 2010; Galanter 1974), upholding truth claims in court has also historically served as a political resource. US torture practices during the country's "war on terror", for example, were not stopped by elected officials or the American public, but human rights lawyers (Hajjar 2023). And as has been argued for the case of Palestine, law is politics and its meaning and application depend on the political intervention of states and people alike (Erakat 2019).

Despite its political potential, the effectiveness of law in general and human rights in particular is limited by its inherent assumptions and institutions. Human rights law, for example, can challenge border violence, but not abolish borders themselves. For example, the

European Convention on Human Rights proscribes that any person has a right to life, as well as the right to the right to be protected from being returned to a place where there the person could face inhumane or degrading treatment. These rights have the powerful potential to make borders more humane, but they do not question the right of states to control entry into their territories. Also, the asylum system creates artificial distinctions between migrants and refugees (Akoka 2020), and drawing on human rights when tackling border violence implies an implicit acceptance of these moral economies of deservingness (Keady-Tabbal and Mann 2021).

In addition, the architecture of human rights relies on states to act in the spirit of signed conventions. The human rights system created in the aftermath of the Second World War offers pathways and tools to lean into state sovereignty, but does no fundamentally challenge or seek to replace the existing nation-state system. At the end of the day, international human rights adjudicating bodies can pass judgments about state practices that violate rights, but the supervision of the implementation of judgments depends again on the states' good faith to implement the judgements.

Despite inherent normative and institutional limits, however, the human rights framework is a resource to reclaim for migrants what Arendt has coined the "right to have rights" (Arendt 1973). In a quest to render legible the violent nature of border regimes, scholars have described border zones as spaces of exception (Agamben 2005), resulting in experiences of 'lawlessnness' (Khosravi 2011, 27). Yet, it is precisely in spaces where people struggle to access their rights that legal and political processes can be mobilized strategically to reduce gaps between human rights norms and actual state practices. What is at stake here is nothing less than migrants' right to have rights. As an Arabic dictum states, a right will not die as long there is a rights-claimer asking for access to this right.

Mobilizing law as a political resource requires approaching law not as a magical tool, but as a complex set of knowledge practices. Unpacking law requires knowledge not only of legal norms and judgments, but also of how a diverse set of legal and political procedures play out practically within and between different institutions. Legislators pass national laws and governments sign international conventions. Judges at the European Court of Human Rights (ECtHR, or the Court) pronounce themselves on rights violations and issue corrective measures to member states of the Council of Europe for both individual applicants and general legal practices. The Council of Ministers and clerks at the Council of Europe oversee the implementation of ECtHR judgments at the domestic level.

Because of its own internal multiplicity, law is able to operate both within and against the state. Shaped by states themselves, law is both fundamentally problematic, but also strategically well-placed to result in effective and tangible changes in the short and medium term for refugees and other migrants. Human rights judgments, for example, can force governments to implement changes in law and bureaucratic and police practices. In the Protocol of the Convention on Torture, for example, states agreed to open up places of detention for regular inspections by independent bodies. Reports on inhumane and degrading treatment of people in detention have directly changed detention practices and thus contributed to resisting torture.[3] In addition, rights violations in court can serve as a legitimizing device for

---

[3] https://www.apt.ch/news/20-years-opcat-still-revolutionising-fight-against-torture

justice struggles of civil society actors. Even when the direct policy impact of court rulings on migrants' rights was limited in the immediate, for example, political actors have amplified the implications of rulings through follow-up advocacy and campaigning (Bonjour 2016; Kawar 2011).

Civil society actors are not alone to approach law in a strategic and instrumental manners, however. After the legal victory for migrants at high sea in Hirsii Jamaa and Others v Italy (2012), for example, European states have learned their lessons from the judgement and turned to adjusting border practices and policies to avoid future scrutiny by the ECtHR (Greenberg 2021). Also, Frontex has succeeded through a set of managerial techniques to erase human rights violations from so-called 'serious incident reports' (Keady-Tabbal and Mann 2023). Reduced to a mere management technique, human rights monitoring is void and risks legitimating questionable state practices.

As human rights can at times be considered not just ineffective, but potentially also harmful (Perugini and Gordon 2015), it requires knowledge alliances across disciplines and sectors to identify when and how to strategically mobilize law, as well as when and how to complement it also with other political resources.

Mobilizing epistemic resources to play legal games:
In my own research practice, I approach law as a knowledge practice that allows me to play a set of games that challenge violent state practices. Drawing on Bourdieu (Bourdieu and Bourdieu 2002), I understand games as a set of different strategies to conserve or subvert the hegemonic rules in a social-spatial arena in which people pursue desirable resources.

In studying and playing different legal games, I try to carve out a space between post-modernist critiques of truth and positivist notions of evidence. I adopt a social-constructivist approach when I study the evidentiary regimes of legal and political practitioners as empirical objects of analysis. As an anthropologist, for example, I researched in collaboration with a legal scholar how court registry staff filter evidence and how judges analyse submitted evidence. By taking seriously on equal grounds the construction of truths outside and inside a court room, the article is able to highlight the political dimensions of seemingly merely technical and legal procedures (Alpes and Baranowksa forthcoming). This is a fundamentally social-constructivist approach to evidence. This article is critical, but not actionable. I thus chose to go two steps further. I mobilized my insights into the politics of legal fact to both play according to the rules and to seek to changes the rules.

On the one hand, I chose to seek a constructive way to try and influence in a positive direction the rules of the game of the Court's evidentiary regime. Together with two legal scholars, we co-wrote an academic article about tools within existing frameworks that allow for improvements in the Court's evidentiary regime. Elaborating this constructive critique required us to take seriously the positionality of judges, as well as the given institutional constraints of a human rights court. In order to be able to make this argument, I also needed to accept the judges' hierarchy between different knowledge forms. To accept this hierarchy does not mean that I agree with it. It means I am willing and able to construct a critique that starts with reality as it is and not as I wish it was. By deliberately adopting a conservative stance, our review of the Court's case law and procedural frameworks demonstrates that ECtHR judges are able to make pushback facts visible by mobilizing procedural possibilities that already exist

(Baranwoska, Alpes and Kienzle forthcoming). The written output takes the form of an academic article in an international peer-reviewed journal. The article is critical, actionable and will count as an academic publication.

On the other hand, I chose to produce and submit evidence to the Court. Playing according to the rules of the game required me to suspend momentarily my more fundamental critique of the human rights system as a whole, as well as my own moral judgment about the biases that I had identified earlier in how different types of evidence are weighed by judges. The written output took the form of a Third-Party Intervention (a written submission to support a pending ECtHR case)[4] and a Rule Nine submission (a written communication to support the implementation of an ECtHR judgment).[5] Within law faculties, such outputs form part of the academic contribution of scholars. As an anthropologist, these pieces of critical and actionable analysis will not count as academic outputs. Yet, a temporary suspension of my doubts and a more fundamental critique allowed me to be actionable and critical in a constructive, even if limited manner. The Polish government has been obliged to respond to our critiques in its implementation of an ECtHR pushback judgment.[6] The ECtHR judges will have to read and consider our critiques of the European border regime when writing the forthcoming judgment for MA and ZR v Cyprus.

In sum, evidence can be a political resource if as scholars we mobilize our epistemic resources for political and legal processes while simultaneously also reflecting on ethical, political and practical implications. Self-reflexivity is crucial in order to navigate whether, when and how to engage with law and evidence as political resources. When, for example, does an aspiration for justice limit the questions we as scholars allow ourselves to ask? In order to remain critical, research design needs to make space for confronting questions about assumptions in our own value system, too.

**Research design principles for transformative knowledge encounters**
"Violence does not persist due to a lack of arresting stories ... but because those stories do not count" (Davies 2022, 3). There is a wide gap today between stories that migration scholars have heard and try to amplify and the stories that drive decision-making in law and policy (Baldwin-Edwards, Blitz, and Crawley 2019) (Orsini 2020). This gap can partly be explained by the hierarchy between different knowledge forms when examined for their potential to influence legal and political processes. Transformative knowledge encounters are based on an understanding how these hierarchies operate, as well as on strategies to challenge and transform them. Whose voices are not heard? Why are some stories not considered to be relevant? Why are some stories not validated as truthful? In a quest for transformative knowledge encounters, I propose three research design principles for critical and actionable migration research: First, build knowledge alliances with justice actors. Second, theorize research needs for justice claims. Third, broker the validity of truth claims. Research design can be fully or partially based on these principles, as well as on only one or all of them.

---

[4] https://hrc.ugent.be/wp-content/uploads/2022/11/TPI_MA-and-ZR-v-Cyprus_logos.pdf

[5] https://hudoc.exec.coe.int/eng#{%22execidentifier%22:[%22DH-DD(2023)153E%22]}

[6] https://hudoc.exec.coe.int/eng#{%22execidentifier%22:[%22DH-DD(2023)212E%22]}    and
https://search.coe.int/cm/Pages/result_details.aspx?ObjectID=0900001680a6cd05

Building knowledge alliances with justice actors: Who needs knowledge?

Knowledge alliances of migration scholars with justice actors pave the way for more transformative knowledge encounters by embracing more horizontal ways of creating, circulating and transforming knowledge practices. In innovative knowledge alliances, migration scholars and justice actors can confront each other to work towards better analytics of what is possible and desirable in the short and medium term. This in only possible if there is trust that all members of the alliance are actually genuinely committed to justice and knowledge.

In order to overcome the deadlock between unactionable resistance and uncritical impact, a diverse set of knowledge practitioners with different types of critiques and positionalities are necessary. In academic discussions of how academics have taken an active role in the world of think tanks and policy advice (Lacroix, Potot, and Schmoll 2021), judges, lawyers and human rights advocacy organizations are missing as potential knowledge users and practitioners. Human rights and refugee rights NGOs and parliamentarians, however, require 'experts' as much as governments do.[7] Other scholars have built knowledge alliances with social workers[8] or refugee-led organizations.[9] These knowledge alliances connect lived experiences with asylum law, as well as migration and social policy and practice. In doing so, these knowledge alliances expand the range of knowledge encounters for academics beyond on the one hand grassroots initiatives and on the other hand state actors and technocratic experts.

A knowledge alliance is only innovative if it allows safe spaces for internal confrontations. When talking about collaborations, it is hence important to keep an eye on how power rhymes with access and critique (Binder et al. 2013, 46–47). The space for internal confrontations is smaller, for example, in alliances with powerful institutions able to grant or block access to fieldwork sites. International organizations and states seek to self-legitimize themselves through knowledge production, which can include the work of monitoring and evaluation agencies (Welfens and Bonjour 2023; Korneev 2018; Kluczewska 2020).

In transformative knowledge encounters, practical knowledge needs to be challenged (Bakewell 2008), as much as it needs to be accounted for. It is in such intellectual and political confrontations that different actors in knowledge alliances can hold each other accountable for when and how it is fruitful or not to engage with what legal and political processes. When, for example, does a focus on the short and medium-term risk replicating what is foregrounded by politics and migration policies, but maybe less central to more long-term justice efforts? Holding each other accountable also requires being able to raise uncomfortable questions. How to reconcile, for example, the everyday needs of people on the move with grassroot aspirations for a world without borders? How to account for possible counter-reactions by states in the

---

[7] My explorations over the last ten years have illustrated that knowledge alliances between an anthropologist and human rights advocates and litigators are possible. For example, empirical data about the implementation of laws can be brought into dialogue with legal rights (Alpes and Majcher 2020; Alpes and et al 2017), and an empirical approach to working of law itself can unravel internal contradictions and avenues to mitigate these (Alpes and Baranowska forthcoming, Baranowska, Alpes and Kienzle forthcoming).

[8] https://www.craftingresilience.nl/research/

[9] https://www.engagedscholarshipnarrativesofchange.org/our-cocreations/refugee-academy

long-run to proposed actions in the short and medium-term? And when is it strategic to accept a set agenda, and when and how is it possible to set your own agenda of issues that should be relevant for policy developments?

Knowledge alliances can choose to blur boundaries between academia and practice either visibly in public or more discreetly behind closed doors. Knowledge alliances can also choose to blur boundaries in practice and perform boundaries in public events. When invited to speak in parliament, for example, I will speak as a scholar whose voice is to count because of my academic training and the methodological rigor, which peers will hold me accountable. Whilst I perform a boundary between academia and the civil society organization that invited me, I am at ease to be publicly seen in this knowledge alliance in the European context. When doing fieldwork in Lebanon, however, I need to also account for my need to be able to access and maintain a legal right to be in the country. Here, I prefer to blur boundaries with human rights organizations more discreetly behind the scene. My support to human right defenders in Lebanon will be more sustainable if I also consider my access to a legal status in the country.

Knowledge alliances require a desire to collaborate beyond the time span of specific projects, as well as generosity and humility. If critically reflected on, knowledge alliances and a willingness to shift between action and inquiry not only paves the way to critical and actionable migration research, but also to deeper learning. Prior to me starting a three-year contract in an ERC-project entitled "DISSECT: evidence in international human rights adjudication", for example, an opportunity arose. Through networking, I had obtained the contact details of 26 survivors of pushbacks from Cyprus to Lebanon. The submission deadline for complaints against Cyprus in front of the European Court of Human Rights had not yet passed. I wanted to interview these survivors in view of potentially submitting new applications to the Court. ERC funding for this research activity was denied as this, I was told, was not academic research. Luckily, EuroMed Rights was able to channel a small budget for these interviews into my hands.

Following the paper trails of the ensuing 26 ECtHR submissions was in the end my most unique and best fieldwork data for the ERC project. Through mere interviews or even participant observation after the official start of my contract, I would have never been able to develop a thick description of the intricacies of the erasure of pushback evidence from the Court. The epistemic violence of law become tangible to me only because I had taken a position and actively participated in the submissions. Yet, NGO money had paid for my fieldwork data. In my experience here, blurring the boundaries between practice and academia has been less an act of giving, than of receiving. Who at the end of the day really helps whom? The answer to this question will not necessarily be clear in advance and points again to the need for more long-term collaborations.

Knowledge alliances can be based on both the co-creation of knowledge and a clear division of labour, recognizing and building on respectively different skill sets. In 2015, for example, I started being in conversation about post-deportation risks with an umbrella organization for undocumented migrants in Brussels, called PICUM. We met and crossed paths over the years, exploring together whether and how research with people after readmissions to countries of nationality could be connected with advocacy against problematic dispositions in the EU return directive. Whilst mind-opening, our conversations did not result in a shared project. When I got funding in 2018 for a research project with deportees from the EU in

Nigeria and Mali, however, I proposed a data sharing agreement. Prior to fieldwork, PICUM's advocacy officer asked me to include additional questions on pre-removal detention into my questionnaires. During fieldwork, I asked research participants whether they agreed for their interview scripts to be shared with PICUM. After fieldwork, I shared the relevant interview scripts, which were only a small amount of my overall fieldwork data. The advocacy and communication officer at PICUM were then in charge of writing up the interview material in a way that they knew would be potentially best received by EU parliamentarians and staff at the Commission. My role was limited to fact-checking and final validation. Designed and produced by PICUM itself, the resulting brochure responded to their need for communication material for their advocacy work than anything I would have been able to draft.[10]

In sum, knowledge alliances with justice actors can support justice claims in a context of post-truth politics, while simultaneously also generating deeper insights for social science research. This, however, requires epistemic modesty, generosity and flexibility.

Theorizing knowledge needs in justice claims: What is relevant knowledge?
Theorizing knowledge needs for justice claims enables more transformative knowledge encounters by explicitly defining what research is relevant for a given political or legal process. The question of relevance cannot be presupposed, but requires explicit and separate research. In order to understand what knowledge is relevant, scholars need to understand both the political and legal processes they want to transform, as well as the role of knowledge in these processes.

Not all knowledge is equally relevant. Knowledge can be critical, but immune to being converted into evidence for justice claims. My book Abroad at Any Cost', for example, is critical of the legal paradigm of trafficking, but the proposed alternative frame of 'migration brokers' in cross-border facilitation processes is not actionable in the short and medium terms for policy makers. Consequently, the book had next to zero impact on policy debates on trafficking and smuggling. It was relevant only to academics who care about emic perceptions of migration risks and illegality. There is politics to story-telling, but an archive of stories and documentation does not without additional work turn into audible justice claims in the corridors of policy makers and judges. What new pathways for actions emerge once academic research has reframed the problem and its causes at hand? This is typically where academics stop and civil society is not able to start.

Determining what knowledge is relevant requires careful co-design with practitioners. Even when working as a consultant at the Dutch section of Amnesty International, I failed to fully draw on existing expertise for research design and consequently most of my research findings were actionable, but not relevant in the specifically available advocacy windows. I had conducted research about post-deportation risks with Congolese deportees in Kinshasa. The findings about detention, enforced disappearances and statelessness pointed to the responsibilities of Congolese state officials and this regardless of respective details in how deportations were implemented by deporting states. Most of the Congolese deportees I had interviewed had been deported by the UK. As the report was going to be published by Amnesty Netherlands, our report needed to speak to Dutch policy makers and government officials. In

---

[10] https://picum.org/wp-content/uploads/2020/10/Removed-stories-EN.pdf

discussion with the respective advocacy officers, we had to come to the conclusion that Dutch policy makers and practitioners would not question their legislation, removal decisions and/or deportation proceedings on the basis of research findings for individuals deported from other countries than the Netherlands. We thus disregarded most of the data in the final report.[11]

Conversations and engagements with human rights advocates elsewhere, however, encouraged me to go further. With permission from Amnesty Netherlands, I published a piece on post-deportation risks in the DRC with the Migration Policy Institute in my own name (Alpes 2019). The publication substantiated prior research findings about risks of detention and torture by a British NGO. In particular, the piece highlighted both real and perceived political activism as a catalyst for detention and extortion, risks beyond the initial detention period at the airport, as well as the importance of identity documents for the life trajectories and respective vulnerabilities of deportees. The British Home Office cited the publication 23 times in their revised policy for Congolese asylum seekers in the UK,[12] and a British law firm asked me to write up an expert report for the appeal of one of their Congolese clients. Findings could be converted into considering whether removal orders were well-founded, as well as into adjusting deportation proceedings and post-arrival assistance. Because of a direct link between empirical research findings and a legal state obligation, the findings were more actionable than those of my book.

In sum, supporting justice claims requires theoretical thinking about the relevance of knowledge about and in legal and political processes. With regards to knowledge about legal and political processes, advocacy and communication officers at human and refugee rights organizations have specialized knowledge about the potential windows of opportunities they represent. With regards to knowledge in these processes, relevant research data needs to be converted into audible knowledge forms at specific moments in time. Unlike in translation processes, only some data will be relevant as evidence. When only minimal parts of the data emerge to be relevant, restricting scholarship to research on convertible data would constitute another form of epistemic violence. Nonetheless, when the story of only one person becomes relevant for a complaint against a state in front of the European Court of Human Rights, then judges are forced to read this person's testimony and to take a decision on whether this person's human rights had indeed been violated.

Brokering the validity of truth claims: How does knowledge come to be validated?
Brokering the validity of truth claims of survivors of human rights violations can facilitate transformative knowledge encounters by intervening in the erasure of voices in legal and political processes. Legal and political processes operate with hierarchies of validity between different knowledge forms in their decision-making processes. In order to identify how to broker the validity of truth claims of those faced with injustices, scholars need to research how knowledge comes to be (in)validated in the respectively targeted legal or political process.

---

[11] https://www.amnesty.nl/content/uploads/2017/07/Rapport-Uitgezet-Mensenrechten-in-het-kader-van-Gedwongen-Terugkeer-en-Vertrek.pdf?x79902
[12] Country Policy and Information Note, Democratic Republic of Congo: Unsuccessful asylum seekers (January 2020). https://www.ecoi.net/en/file/local/2023159/DRC_-_CPIN_-_UAS_-_v4_-_final.pdf

Legal rules of evidence and academic standards of validity and probability do not necessarily meet (Loevinger 1992). Whilst academia validates knowledge through peer-review, judges in Court and officers in asylum procedures assess evidence according to their own standards of proof. If validity is a social and not an epistemic accomplishment (Aradau and Huysmans 2019), then scholars need to broker the validity of truth claims based on insights into how knowledge comes to be validated by respectively relevant practitioners in position of authority in legal and political processes. Drawing on the anthropology of development, I understand brokers as go-betweens or intermediaries who occupy a strategic function in mediating tensions, for example, between global structures and local actors (Olivier de Sardan and Olivier de Sardan 2005).

Knowledge brokers in legal and political processes need to pay attention to the analysis of evidence, not just its mere presence or submission. At the European Court of Human Rights, for example, applicants generally have the burden of proof, meaning that applicants need to prove that a rights violation has taken place. Under specific circumstances, however, the Court can shift the burden of proof to the state, which then means that it is on states to refute the allegations raised by the applicants. One of many counter-strategies to the erasure of voices can be a set of legal arguments for the shifting of the burden of proof (Baranowska 2023).

Different justice claims require insights into the validation processes of different actors. For policy recommendations in the humanitarian sector, for example, the last decade has seen a turn by funders to request the so-called "community-validation" of research results. Here, participatory research methods can transform how and what knowledge is created, basing thus validity and any ensuing truth claim on the participation of research participants themselves (Alpes et al 2023, Davids, van Houte and Alpes forthcoming).

When seeking to make pushbacks visible in judgments, by contrast, it is important to present knowledge in forms that have a high chance to be accepted as valid by judges. Between 2001 and 2024, I researched how the voices of pushback survivors come to be (in)validated at the ECtHR. My deepest insights came from my biggest failures. Out of 27 complaints against Cyprus based on interviews I had conducted with survivors, the ECtHR's registry office registered only one. The non-registration letters stated that "there (was) no official confirmation of the alleged government action having taken place." The registry staff mobilized this as a reason to not register the complaints although the applicants' lawyer had explained that the pushback survivors had "never (been) notified with an entry ban/deportation/expulsion decision in writing explaining the reasons in fact and in law", nor had any "record (been) made of (their) wish to apply for asylum."

As the Cypriot coastguard officers had acted in complete informality, it was not possible for the lawyer to submit official proof of an unofficial practice. The applications were not registered because documents that did not exist were not submitted. The one application that the ECtHR registry registered was a case for which the lawyer had been able to file an interim measure, which is an emergency measure to put a halt on state action so as to prevent human rights violations. While the interim measure had not succeeded in preventing the pushback, it had forced the Cypriot state to go on record in response to questions asked by the ECtHR. Only here did the lawyer have official proof of an unofficial practice. ECtHR judges give greater weight to state-produced evidence than to the voices of people.

Insights into the assumptions that drive the knowledge politics of judges can allow scholars and civil society organizations to broker the validity of truth claims of survivors of human rights violations in slightly different ways. Based on the Cyprus case, for example, I have concluded that it can be more strategic to mobilize law to produce knowledge, than to produce knowledge to inform the law. Legal procedures to produce knowledge can include not only interim measures, but also freedom of information requests and investigations into states' positive obligations to record practices, such as the deprivation of liberty and respect for procedural safeguards. Also, criminal proceedings at the domestic level (Fehr and Alpes 2024) and infringement proceedings before the Court of Justice of the EU can help in gathering evidence on the existence of breaches of fundamental rights that stands a high chance of coming to be validated as proof.[13]

This lesson raises questions about the role of social scientist in brokering the validity of truth claims about injustices in legal and political processes. For the case of pushbacks, several options come to my mind. First, empirical insights can support legal strategies. For example, empirical insights about surveillance material can be useful information for subsequent freedom of information requests. Second, empirical research can be critical to document the unwillingness of some state actors to produce evidence, or to avoid legal scrutiny (Majcher and Strik 2021). Third, social scientists can bring new empirical data to the analysis of legal processes, which can help to both identify bias and avenues for improvements.

In sum, supporting justice claims requires scholars to understand and relate in one way or the other to the validation practices in the respectively targeted legal or political processes. When scholars chose to broker the validity of truth claims of survivors of human rights violations, this means neither that they approve of how knowledge comes to be validated in respective legal and political processes, nor that they limit themselves to knowledge production able to broker such truth claims. Instead, it is important to explore how different counter-strategies to the erasure of evidence can complement each other in the mid and long-term (Herremans 2023).

**Conclusion:**

This article has examined the circulation and confrontation of different knowledge practices and forms from the lens of how scholars can mobilize their epistemic resources to build symbolic capital for justice claims. Drawing on experiences as policy officer and research consultant at human and refugee rights organization, as well as an anthropologist at law faculties, the article demonstrated that "impact" is nothing to be "done" once the research is completed. Instead, critical and actionable migration research is based on research design principles that critically studies and constructively engages with the knowledge practices of practitioners.

By thinking about the work that knowledge can do in legal and political processes, the article makes a plea for critical and actionable research as an avenue to participate in the co-creation of "public truths." The article defines critical and actionable migration research as proposing constructive critique that can be translated into concrete actions in political and legal processes in the short and medium term. Based on this definition, the article puts forward that

---

[13] https://www.greens-efa.eu/files/assets/docs/budgetconditionality_study_web_28_pages.pdf

both law and evidence can be mobilized as political resources to mitigate state violence and benefit migrants' rights and dignity. First, evidence can be a trojan horse, used to smuggle uncomfortable, but audible truth claims into legal and political processes. This smuggling act requires awareness and respect for the temporal horizons and possibilities of practitioners in political and legal processes. Second, the rule of law can be a strategy to operationalize resistance to violence. This smuggling act requires a balancing act and a flexible evolution between radical and comprehensive types of critique.

The article proposes three research design principles for transformative knowledge encounters: building knowledge alliances with justice actors, theorizing research needs for justice claims and brokering the validity of truth claims. Based on an understanding of practitioners as situated thinkers, these research design principles allow for strategic thinking across sectorial boundaries on what knowledge is relevant and likely to be accepted as valid in targeted legal and political processes. Reconsidering relevance and validity together with justice actors requires epistemic modesty, flexibility and generosity. As scholars, we do not have to agree with what counts as relevant and valid. Yet, if we want our critique to have impact, then we need to understand the rules of the game, and either play by them or constructively engage to transform them.

Critical and actionable migration research offers possibilities for theorizations about transformative knowledge encounters at a mid-range level of abstraction. Transformative knowledge encounters, for example, conceptually presuppose the existence of knowledge brokers, the conversion of research into different knowledge forms, as well as of analytics of what is possible and how within a range of aspirations for just futures in the short and medium term. There is politics to migration law and policy and the attempt to feed knowledge and truth claims into political and legal processes can be an avenue both for (epistemic) resistance, as well as a process for intellectual insights into knowledge politics itself. The effects of knowledge in legal and political processes are in and of themselves objects of inquiry. Incidentally, theoretical reflections at this mid-range level also allow for new types of partnerships with scholars in the Global South who often navigate the need and pressure for impact from less privileged positions.

In proposing to elaborate a grounded theory of transformative knowledge encounters, I hope this article will encourage more systematic reflections on research impact in migration politics. Fundamental research beyond aspirations for impact is key for long-term transformations, and one of the core privileges of academia is precisely the freedom to develop radical critiques and utopian thinking. If academic pursuit is reduced to practical or activist concerns, then this limits its intellectual depth. The political commitment of scholarship can also create blind spots for uncomfortable questions. If some migration scholars, however, choose to try and mobilize their epistemic resources to engage with justice claims, then this requires a thorough reflection about the conditions that enable academics to broker knowledge in manners that are truly transformative and meaningful. In other words, academics should either be explicit and transparent about their theory for impact, or not do it at all.

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

**Acknowledgements:**

Anything worthwhile is hardly ever achieved in isolation. First of all, I would like to thank the senior editorial fellows of the Migration Politics journal, Saskia Bonjour, Darshan Vigneswaran and Evelien Ersanilli. I would not have written an autobiographical article, had it not been for their encouragement. I am particularly grateful for many repeated rounds of comments from Saskia Bonjour, as well as feedback on early versions of the article by Heath Cabot and Ashwini Vasanthakumar. In the process of working through the many layers of my experiences and its lessons for academic research, I also had the luxury of being in conversation with many different scholars outside of the realm of Migration Politics. Michaela Pelican has allowed me to enter joint critical conversations first in the framework of a seminar at the Global South Studies Centre at the University of Cologne in June 2022 and then at the joint autumn research academy "Dialogue on Migration Governance in the Euro-Mediterranean Region (DiaMiGo) of the American University of Cairo and University of Cologne in October 2023. In July 2023, Saskia Bonjour and I co-organized a workshop at IMIESCO entitled "Knowledge politics: scholarship and 'evidence' in the dangerous politics of migration." I would like to thank Halleh Ghorashi, Maria Shaidrova, Robert Vandervoort, Marta Welander and the conference participants for having joined the workshop. In November 2023, the "anthrostate" group of the European Association of Sociologists provided me with an opportunity to present and discuss work in progress. In January 2024, Nora Stel invited to me to present the paper at the 'Revisiting Interdisciplinary Migration Studies' seminar series at Radboud University and allowed me to receive feedback from Tineke Strik. Outside of these formal realms, I am grateful for conversations and feedback with Katharina Natter, Nora Stel, Anouk de Koning and Nassim Majidi. Finally, my thanks also go to the Human Rights Research Network and Cessmir for having co-organized a workshop and a roundtable in April 2024 at Ghent University around the key topics of the article. Here, I am thankful to comments from above all Giacomo Orsini and Brigitte Herremans, but also Marika Sosnowski, Ine Lietart, Ellen Desmet and other present Cessmir researchers. My writing time was supported by the Horizon 2020 research project 'DISSECT: Evidence in International Human Rights Adjudication', funded by the European Research Council through an Advanced Grant (ERC-AdvG-2018-834044) and led by Marie-Benedicte Dembour.