# Peer review of "Smuggling critique into impact: Research design principles for critical and actionable migration research"

_Migration Politics_

## Round 1 · Referee Report · Anonymous (Referee 1) · 2024-7-23

Report
The article discusses how academics can leverage their knowledge to challenge border violence and human rights violations against migrants. Despite increased research, border violence persists due to policymakers ignoring critical but impractical recommendations and actionable suggestions failing to improve migrants' conditions. It seeks the “conditions of possibility for transformative knowledge encounters” (p. 2) and suggests that migration scholars should increase the potential for impact not my producing more evidence but treat law as a knowledge practice. Moreover, they should adjust the research design phase towards three principles: To build knowledge alliances with justice actors, to theorize knowledge needs in justice claims, and to broker the validity of truth claims. The article’s aim is to show that “resistance and impact do not have to be fundamental opposites” (p. 6).
The article aligns well with the overarching goals of Migration Politics, as it bridges the gap between anthropological research experiences and political science studies on the use of evidence in migration policy, its application, and migration litigation. It employs clear and accessible language, making its significant and convincing argument readily comprehensible. The acceptance criteria are met, namely the expectation to open a new pathway in an existing or a new research direction, with clear potential for multi-pronged follow-up work, as well as the general acceptance criteria. However, the author should consider the following points before publication.
Despite its quality, the paper appears to engage with existing debates in anthropology and the social sciences on the use of evidence without explicitly referencing these discussions. Both the second and third research design principles, along with the example on page 10, address the challenges faced by academics, particularly anthropologists, when providing evidence in contested contexts such as migration policy or litigation. Research design principle two discusses the necessity for academics to define the relevance of their knowledge for specific contexts in advance. Research design principle three examines how academics can support the validation of evidence when it is applied outside of academia. The example on page 10 illustrates how the author produces evidence for the European Court of Human Rights (ECtHR). However, there is a wealth of reflexive research on these issues, notably from anthropologists acting as expert witnesses in asylum procedures.
For example, scholars discuss the trade-off between providing "useful" (actionable) knowledge in asylum procedures to support protection claims and the constant threat of being misunderstood by decision-makers (e. g. Murray 2017). Anthony Good (2007) and others reflect on the problem that anthropologists must simplify the complexity of their knowledge to fit the binary decision-making process in refugee status determination (protection/no protection). This aligns with the author's observation that "insights into the assumptions that drive the knowledge politics of judges can allow scholars and civil society organizations to broker the validity of truth claims of survivors of human rights violations in slightly different ways" (when binary decision-making is seen as an example of these assumptions). Markus Höhne (2016) writes about his approach of "strategic essentialism," which enables him to anticipate context-specific legal requirements and assumptions by working with categories and assessments in expert opinions that he would otherwise avoid. Additionally, Levy (2022) and McDonald-Norman (2014) describe how decision-makers give greater weight to state-produced evidence, an observation the author also makes regarding the ECtHR.
In its current form, the author almost exclusively relies on autobiographical explorations. These could be expanded and enriched by incorporating the observations and reflections from a broader range of colleagues. The paper should acknowledge that many of these debates are already ongoing, with similar theses and results. Its indisputable quality lies in adding another voice to these discussions and theorizing them through the three research design principles.
Another issue concerns the metaphor of “smuggling” critique into impact. This term imparts a dubious connotation to the paper's legitimate claims, potentially undermining their strength. Additionally, it appears to contradict the paper's concluding assertion that academics should be explicit and transparent about their theories for impact. The notion of “smuggling” critique seems at odds with the requirement for transparency and explicitness.
Please find some minor points below:
-
“Despite increased focus on research impact, however, border violence is only increasing.” Whose focus does the sentence refer to? Of policy-makers? Or of the scholars?
-
“Also, the asylum system creates artificial distinctions between migrants and refugees (Akoka 2020), and drawing on human rights when tackling border violence implies an implicit acceptance of these moral economies of deservingness (Keady-Tabbal and Mann 2021). Why? The non-refoulement principle based on Art 3 (legally) counts for everybody. The moral economy of deservingness is indeed central for the EU asylum regime, but at this point, there is no underlying (legal) assumption that some migrants deserve refoulment and others don not.
-
On p. 9, the litigation on push backs is the example for the mobilization of rights However, the example lacks to mention more recent and more restrictive ECtHR litigation on that matter (namely ND/NT)
-
The application of a social-constructivist approach (p. 6, p. 7, p. 9) is often stated but never defined.
-
“Violence in Gaza and transitional justice efforts in Syria, for example, are also marked by epistemic violence that begs for a critical reflection on responsible scholarship.” In how far? This sentence requires explanation.
-
“Migration studies today is marked by a lack of dialogue between conventional scholars basing themselves on Weber and critical scholars basing themselves on Marx and Foucault (Favell 2022). This sentence also requires explanation. It sounds sophisticated, but at least one or two sentences on in how far (all?) conventional scholars rely on Weber and (all?) critical scholars on Marx and Foucault would be helpful.
References used above:
Good, Anthony (2007): Anthropology and Expertise in the Asylum Court, New York Höhne, Markus V. (2016): The strategic use of epistemological positions in a power-laden arena: anthropological expertise in asylum cases in the UK, International Journal of Law in Context 12 (03), 253-271. Levy, Jordan (2022): Competing Versions of Reality in Honduras: State Theory as a Tool for the Anthropologist Expert Witness, Annals of Anthropological Practice 46 (1), 83-86. McDonald-Norman, Douglas (2014): Simply Impossible: Plausibility Assessment in Ref-ugee Status Determination, Alternative Law Journal 39 (4), 451-455. Murray, David A.B. (2014): Real Queer: “Authentic” LGBT Refugee Claimants and Homonationalism in the Canadian Refugee System, Anthropologica 56 (1), 21-32.
Requested changes
1. Connecting the discussion with the anthropological debate on the use of expert knowledge especially in refugee status determination
2. Reflect on the use of the term “smuggling” and include at least some considerations why this metaphor has conceptual added value.
3. See comments on “minor points”, especially the requirement of a definition of “social-constructivist approach”
Recommendation
Ask for minor revision

---

## Round 1 · Referee Report · Anonymous (Referee 2) · 2024-7-29

Report
This article undertakes the very worthwhile endeavour to explore the possibilities for finding a middle ground between non-pragmatic critical migration research and non-critical pragmatic policy-oriented approaches in order to influence current migration policy. While in the second part of the paper, where three research design principles are proposed, the argument is already quite convincing, in my opinion it is ‚not there yet‘ in the first parts. The main difference between these parts is a) how terms are employed, b) the usage of examples and c) the flow of the argument. Concerning a) terms, there are many of them employed in the paper and only some of them are discussed in more detail. Others are not, even though they seem highly relevant for the paper, such as ‚epistemic resources‘, ‚justice (claims)‘, ‚truth‘ or ‚evidence‘, and it does not really become clear on which grounds concepts are discussed (or not). When an explanation is given, there is a tendency that ‚big concepts‘ are used, which however sometimes open more questions than they answer. To give some examples - on p. 2 ‚critical and actionable migration research‘ is defined by using a quote from Lorenzini and Tazzioli on ‚critique‘ as a contestation of „regimes of truth that govern us“, which would probably also include far-right movements as part of ‚critique‘, but I wonder if the authors means that. On p. 6, ‚damage-centred research‘ is mentioned without explanation, ‚epistemic violence‘ is discussed very briefly, the example given (concerning European borders) remains somewhat opaque and could be spelled out more clearly (what exactly is meant by ‚practices of denialism‘), and the discussion on ‚post-truth‘ politics is again only touched on in a rather short paragraph. Also here, some bigger questions arise – what does the violence stand for in ‚epistemic violence‘? Does is stand for ‚domination‘, ‚power‘, ‚injustice‘, ‚distortion‘? Is ‚suppression‘ also referring to for instance religious ways of knowing? What is meant by evidence? How can one gain evidence, who/what is believed? What is seen as truth? These bigger concepts/terms are, however, not discussed in the paper. Evidence, for instance, seems to be a main term for this article, yet on p. 10 I was still not sure how it was used. It is not a self-evident (sic) term, as the discussions around it in Science and Technology Studies show. I was also surprised that Boltanski and Thévenot were not mentioned at all, as they have written about many of the issues discussed in the paper (critique, justification, economies of worth) This myriad of terms and concepts appears especially on pp. 5-10, where the author, even though talking about exploring a middle ground, somehow gets stuck in (‚critical‘) theory as well. My actionable proposition here would be to use less of the ‚big terms/concepts‘ and take more time for discussing the ones used. In addition, using more b) examples would clarify the argument as well. Furthermore, the examples used should be as much contextualised and as clear as possible. In the first part of the article, only few examples are given in the first part, and those given are mainly very brief. The first real example is given on p. 7/8 and immediately helps to understand the argument; another example on detention practices is given on p. 8, which however could be even more precise (when/where were detention practices changed due to reports?); the case mentioned on p. 9 (Hirsii Jamaa vs Italy) should be explained in one sentence for non-insiders; on the same page it is mentioned that human rights can potentially be harmful, also this would need an explanation. From p. 10 onwards there are more examples given and those given are clearer. On p. 15 the example on shifting the burden of proof as counter-strategy to the erasure of voices, could maybe be even more explicit? What difference does the shifting of burden of proof make? What kind of effect does it have? Focussing on less terms and giving more examples would also clarify the c) argument of the paper, especially in the first part. It would be good to make clearer from the beginning what the article intends to do and here the first sentence from the last paragraph (p. 17) could come earlier. A better flow of the argument would also help to erase some repetitions of ideas and half sentences. There are two more aspects I would invite the author to think about in terms of the argument – the dichotomy of the two ‚ideal types‘ of research and the arenas of action discussed in the article. The argument builds strongly on a dichotomy between non-critical applied research and non-actionable critical research. Maybe it would be good to treat these two approaches as ‚ideal types‘ in the Weberian sense. Within the paper, there are many terms aligned to them: to ‚critical‘ research –> Marx/Foucault, resistance, theory-driven, oriented towards migrant voices, no-border movement, radical, destructive, post-modernist critiques of truth; to ‚conventional‘ research – Weber, policy-driven, oriented towards state actors, comprehensive, positivist, impact, positivist notions of evidence. Looking at these terms, I was wondering if there wasn’t some sort of implicit judgment in the description of the two perspectives. I was also wondering about the argument on p. 5 (apparently borrowed from Favell), that distinguishes between Weberian ‚conventional‘ and Marxian ‚critical‘ scholars – is ‚conventional‘ the right term (this seems to be one of the terms with a moral judgement)? Is this really about Weber vs. Marx, and if yes, in terms of research perspective (where Weber also builds on Marx)? Or about the role research plays for politics? Where would you categorize Bourdieu then, who tried to bring both together? And is the differentiation into these different perspectives really a result of how different forms of research are used (or not used) by policy makers (as the abstract reads)? Concerning the arena of action presented in the article, the author describes the text as being about knowledge in legal and political processes. The examples, however, focus on the arena of court cases, while other arenas, such as making or changing of laws or policies are alluded to in the article, but do not play a major role in the examples. For clarity, it might be better to restrict the discussion to court cases as one arena in the bigger field of legal and political processes. I suggest this also because evidence might work differently in other arenas, for instance academic statements concerning proposed changes of migration/refugee law. In my experience, this presupposes again a different way of presenting testimony/evidence than what is described in this paper. This would also clarify the argument again, as for instance on p. 15 the short paragraph about policy recommendation seems a bit out of place. This focus on the arena of the court could also help to answer a questions that came to my mind very early when reading the paper: is the transfer of research knowledge into policies really ‚just‘ a matter of law? And what about researchers not dealing with the field of law – isn’t the article insinuating that they do not have the ability to do what the author does, i.e. to smuggle critique into impact? smaller issues: - p. 2 - the author seeks to open a ‚grey space‘ – I wonder if that is the right term. At least the geographer Oren Yiftachel uses the term quite differently for urban development in Israel when he situates grey spaces between the ‘lightness’ of legality/approval/safety and the ‘darkness’ of eviction/destruction/death. - p. 4 – I am not sure about ‚transitional justice‘ in Syria with the old system still standing (whose justice is that?) - p. 5 – the author claims that the term ‚consultant‘ sounds dirty in the realm of social sciences – this is probably also an issue of elitism, as consultancies are partly also taken by academics with insecure job situations; also on p. 6 these two research perspectives are called ‚aspirations‘ which leaves aside economic insecurity and political pressure to include impact in research proposals for funding - p. 8 – it is argued that strategic mobilization of law takes place in spaces where people struggle to access their rights – this, however, necessitates that there are actors who can mobilize the law, I wonder how that works for instance for women in Afghanistan in terms of accessing education - p. 9 – the arena is not clear here – the author writes about having mobilized insights into the politics of legal fact to do something, but it is not clear what that mobilization refers to – action beyond writing? or to writing this article, or another article? further down the author mentions to have adopted an conservative stance, what does that mean? also this example could be spelled out in a bit more detail (what kind of procedural possibilities do you mean?) - p. 10 - what does ‚MA and ZR v Cyprus‘ mean? - p. 12 – isn’t taking part in a law case participant observation as well? - p. 13/14 – I am wondering in how far this is really about theorising knowledge needs or more about practical learning of these needs - p. 15 – at first the author differentiates between evidence in court and validity in academia, but later talks about validation of knowledge in legal process, this is a bit confusing; in addition ‚broker‘ is defined here, in my view this explanation is not necessary for understanding the argument - p. 17 – the last sentence of the article is pretty harsh - in the reference section, there is in some cases a doubling of names (Bourdieu, de Sardan)
Recommendation
Ask for major revision

---

## Editorial Decision

unknown